# Comparison of Five Assays for the Detection of Anti-dsDNA Antibodies and Their Correlation with Complement Consumption

**DOI:** 10.3390/diagnostics15192430

**Published:** 2025-09-24

**Authors:** Vincent Ricchiuti, Jacob Obney, Brooke Holloway, Mary Ann Aure, Marti Shapiro, Chelsea Bentow, Michael Mahler

**Affiliations:** 1Labcorp North Central Regional Reference Laboratory, Dublin, OH 43016, USA; jacobney23@gmail.com; 2Research and Development, Werfen Autoimmunity, San Diego, CA 92131, USA; bholloway@werfen.com (B.H.); maure@werfen.com (M.A.A.); mshapiro@werfen.com (M.S.); cbentow@werfen.com (C.B.); m.mahler.job@web.de (M.M.)

**Keywords:** SLE, lupus, disease activity, dsDNA, autoantibodies

## Abstract

**Background**: Anti-dsDNA is an important biomarker for the diagnosis, prognosis, and monitoring of systemic lupus erythematosus (SLE). Although several assays for anti-dsDNA antibody detection are routinely used, standardization remains limited, and differences have been reported. This study aimed to compare five methods for anti-dsDNA antibody detection and to estimate their association with complement consumption. **Methods**: A total of 149 samples submitted for routine laboratory testing were collected and tested on five platforms: *Crithidia luciliae* indirect immunofluorescence test (CLIFT), addressable laser bead immunoassay (ALBIA), a high-avidity (HA) enzyme-linked immunosorbent assay (ELISA), chemiluminescent immunoassay (CIA), and a novel particle-based multi-analyte technology (PMAT). Complements C3 and C4 were available for a subset of the total samples. **Results**: Correlation between anti-dsDNA assays ranged from 0.94 (CIA and PMAT) to 0.65 (ALBIA and CLIFT). The AUC from the ROC analysis using CLIFT as a reference was 0.95 for PMAT, 0.94 for CIA, 0.93 for ELISA, and 0.86 for ALBIA. The highest sensitivity relative to CLIFT at a fixed specificity of 94.4% was 84.7% for CIA and ELISA, 76.3% for PMAT, and 42.4% for ALBIA. Correlation between anti-dsDNA and C3 ranged from −0.81 for ELISA to −0.51 for ALBIA. **Conclusions**: Different anti-dsDNA detection methods showed varying diagnostic performance and correlation and varying agreement with CLIFT and complement consumption. ELISA, CIA, and PMAT showed high correlation to each other and to CLIFT and were in strong concordance with low C3 levels. In contrast, ALBIA revealed lower clinical performance and correlation with CLIFT and complement consumption.

## 1. Introduction

Anti-dsDNA (double-stranded DNA) antibodies are highly specific markers and are part of the classification criteria for systemic lupus erythematosus (SLE) [1]. They correlate with disease activity and contribute to the pathogenesis of lupus nephritis (LN). Additionally, anti-dsDNA antibodies can mediate inflammation and tissue damage, leading to a range of symptoms and complications in affected individuals. Furthermore, monitoring of anti-dsDNA antibody levels is valuable for assessing disease progression and guiding treatment in SLE patients. The anti-dsDNA antibodies most relevant for the diagnosis and assessment of disease activity are primarily of the IgG class. Although IgM and IgA antibodies are also present, their clinical relevance remains poorly understood [2,3,4].

Several assays can detect anti-dsDNA antibodies in a clinical setting [5,6]. The choice of assay may depend on the specific laboratory’s capabilities and the clinical context. Common methods include the following (in chronological order of availability):Farr Radioimmunoassay (RIA): This utilizes radiolabeled DNA to detect anti-dsDNA antibodies. It is known for its high sensitivity and specificity, although its use is less common due to concerns about radiation exposure. It detects anti-dsDNA antibodies of all occurring immunoglobulin isotypes: IgG, IgM, and IgA.*Crithidia luciliae* immunofluorescence test (CLIFT): This uses a kinetoplastid parasite, *Crithidia luciliae*, as the substrate for detecting anti-dsDNA antibodies of the IgG isotype. It is known to be highly specific for SLE but offers low sensitivity [7].Enzyme-linked immunosorbent assay (ELISA): This is a widely used technique for measuring the presence and levels of anti-dsDNA antibodies, providing quantitative results that aid in assessing disease activity. Most anti-dsDNA ELISAs are highly sensitive but have limited specificity, as they detect both high- and low-avidity antibodies, even though only high-avidity antibodies are clinically relevant for SLE [8]. Only a few of the commercially available anti-dsDNA ELISAs are designed to compensate for this lack of specificity by using a high salt concentration in the buffers to remove low-avidity antibodies [9]. Most commercial ELISAs detect anti-dsDNA IgG.Fluorescent enzyme immunoassay (FEIA): This method is similar to ELISA but uses fluorescence for the detection of anti-dsDNA IgG antibodies. FEIA is fully automated, offering ease of use; however, it has similar limitations to those of the ELISAs.Addressable laser bead assay (ALBIA): This modern multiplex method enables the simultaneous detection of multiple autoantibodies, including anti-dsDNA IgG, in a single test, and is based on a laser detection method. It can offer a more comprehensive autoimmune profile in one reaction.Chemiluminescence immunoassay (CIA): This modern technology is based on antigen-coated paramagnetic beads and chemiluminescence as the detection method [10]. Due to the stringent washing procedure, made possible through the wash steps with magnetic fixation of the beads, this assay is highly specific for SLE, as most low-avidity antibodies are removed. The antibodies detected are of the IgG class.Particle-based multi-analyte technology (PMAT): This is a novel method for measuring IgG isotype autoantibodies to dsDNA, which has recently become available to laboratories [11]. Similarly to ALBIA, PMAT offers the simultaneous detection of multiple autoantibodies using an advanced and sensitive detection approach based on fluorescence.

In addition to anti-dsDNA antibodies, complement status represents a very important biomarker in SLE and is known to play a role in disease pathogenesis [12]. Complement levels, particularly C3 and C4, are often used to assess disease activity. Complement consumption is linked to organ involvement, especially in lupus nephritis (LN), and can help predict flares and monitor treatment response. Measuring both anti-dsDNA and complement markers can give clinicians a more comprehensive understanding of the SLE patient and the underlying pathogenic mechanisms and a broader assessment of the disease activity and severity.

The choice of assay may depend on factors such as performance characteristics, including sensitivity and specificity, desired level of automation, cost, availability, and the particular clinical scenario. The interpretation of results should always consider clinical symptoms and other diagnostic criteria when assessing autoimmune diseases, particularly SLE. This study aimed to compare five methods for anti-dsDNA antibody detection and to evaluate their association with complement consumption, which served as an indicator of SLE disease activity in the absence of clinical data.

## 2. Materials and Methods

### 2.1. Patient Samples

A total of 149 serum samples were collected at Labcorp as part of routine testing for anti-dsDNA. Of these, 113 were from female patients and 34 from male patients, ranging in age from 12 to 91 years. Demographic information was unavailable for 2 samples.

### 2.2. Measurement of Anti-dsDNA Antibodies

All samples were tested for anti-dsDNA using five out of the seven methods described above, *Crithidia luciliae* (anti-dsDNA) EUROPattern (software version 3.5) (Euroimmun, Lübeck, Germany), QUANTA Lite High Avidity (HA) dsDNA ELISA (Werfen, San Diego, USA), BioPlex 2200 dsDNA (software version 4.4) (ALBIA) (BioRad, Hercules, CA, USA), QUANTA Flash dsDNA (CIA) (Werfen, San Diego, CA, USA), and Aptiva dsDNA (Werfen, San Diego, CA, USA), all in accordance with the manufacturers’ instructions. An overview of the technical characteristics of all anti-dsDNA assays can be found in Table 1. The Farr assay was excluded due to its current limited use.

*CLIFT* is an indirect immunofluorescence (IIF) assay for the screening and titration-based determination of anti-dsDNA antibodies in human serum. The assay employs the hemoflagellate *Crithidia luciliae* as a substrate. This single-cell organism possesses a giant mitochondrion containing a highly condensed mass of circular dsDNA. The QUANTA Lite^®^ HA dsDNA assay is an ELISA designed to preferably detect anti-dsDNA antibodies of high avidity that uses a native antigen from calf thymus. This differentiates it from other anti-dsDNA ELISAs that measure anti-dsDNA antibodies of all avidity classes, from low to high. As a consequence, QUANTA Lite^®^ dsDNA HA is more specific for SLE compared to other ELISAs [13,14]. The BioPlex 2200 dsDNA assay uses a synthetic dsDNA antigen and works with an automated analyzer that uses multiplex bead technology to simultaneously detect antibodies to several antigens in a single tube [15,16]. The principle of the QUANTA Flash anti-dsDNA CIA has been previously described [6] and was developed using synthetic dsDNA coupled to the surface of paramagnetic beads. And finally, the PMAT anti-dsDNA assay, that also leverages synthetic dsDNA as the antigen, is part of a panel for the simultaneous detection of multiple autoantibodies in one single step [15,16]. This technology is based on the use of a mixture of suspended microparticles that are aligned in a monolayer and analyzed through digital imaging technology using two LEDs at the end of the incubation procedure.

### 2.3. Measurement of Complement Consumption

To assess the correlation with disease activity, we used complements C3 and C4 as serological surrogates for this clinical feature and compared each anti-dsDNA assay with these complement markers measured using Cobas^®^ c 502 (Roche Diagnostics, Indianapolis, IN, USA). C3 and C4 data were available for 49 and 48 of the samples, respectively. The laboratory-specific range for C3 is 82–167 mg/dL and for C4 is 14–44 mg/dL.

### 2.4. Statistical Analysis

Receiver operating characteristic (ROC) and area under the curve (AUC) analyses (with 95% confidence intervals) using CLIFT results as a binary classifier were used to assess the concordance independent of the cut-off values [DataLab, Werfen, San Diego, CA, USA].

Visualization of the overlap in positivity for various dsDNA assays was performed using the ‘Upset plots’ library in Python version 3.8, as described by Lex and colleagues [17].

Correlation between assays was visualized using heatmaps [DataLab, Werfen, USA] utilizing Spearman’s correlation analysis.

## 3. Results

### 3.1. Frequency and Intersection of Anti-dsDNA Positivity Across Platforms

Of the 149 samples analyzed, 50 (33.6%) were consistently negative for anti-dsDNA antibodies across all five testing platforms. Among the positive anti-dsDNA results, the highest concordance was observed in 48 samples (32.2%) that tested positive across all five methods, followed by 15 (10.1%) samples positive on PMAT, CIA, and ALBIA. Eleven (7.4%) samples were positive on ALBIA alone, while nine (6.0%) were positive on four methods, namely, PMAT, ALBIA, CLIFT, and CIA. The full distribution and overlap of positive results across methods are illustrated in Figure 1.

### 3.2. Correlation Between Anti-dsDNA Assays

The overall agreement between the different methods was relatively high in our cohort. The quantitative concordance as expressed by Spearman’s *ρ (rho)* coefficient varied between 0.65 (ALBIA vs. CLIFT) and 0.94 (PMAT vs. CIA) (Table 2; Figure 2).

Additionally, the findings were compared to the results derived from CLIFT testing, since CLIFT is often used as the confirmation assay for anti-dsDNA due to its very high specificity. Sensitivity and specificity relative to CLIFT were plotted against each other for all remaining four assays. The highest AUC from the ROC analysis was found for PMAT (0.95), followed by CIA (0.94), ELISA (0.93), and finally ALBIA (0.86) (Figure 3).

Supplementary to the correlation analysis, a different strategy was used to assess and compare the performance of the different anti-dsDNA assays. The specificity was set to a fixed value of 94.4%, and relative sensitivity compared to CLIFT at this specificity was calculated. The highest sensitivity relative to CLIFT at this fixed specificity of 94.4% was found for CIA and ELISA (both 84.7%), followed by PMAT (76.3%) and, with a significantly lower value, for ALBIA (42.4%) (Table 3).

### 3.3. Correlation Between Anti-dsDNA Assays and Complements C3 and C4

C3 data were available for 49/149 (32.9%) of the samples, with C3 levels ranging from 15–200 mg/dL. When analyzing the inverse relationship between C3 as a proxy for disease activity and anti-dsDNA antibodies, an overall negative correlation was found for all of them. The assays ranked in terms of agreement from highest to lowest negative correlation in the following order: ELISA (−0.81), CLIFT (−0.76), CIA and PMAT (both −0.70), and finally ALBIA (−0.51) (Figure 4) (Spearman’s *ρ*: −0.812 to −0.510; *p* ≤ 0.0001).

C4 data, on the other hand, were available for 48/149 (32.2%) of the samples, with C4 levels ranging from 2 to 51 mg/dL. When analyzing C4, the assays ranked in terms of agreement from highest to lowest were similar to C3: ELISA, CLIFT, CIA, PMAT, then ALBIA (Spearman’s *ρ*: −0.654 to −0.327; all *p* < 0.0001 except for ALBIA: *p* = 0.0234). 

## 4. Discussion

Anti-dsDNA antibodies are a hallmark in the diagnosis of SLE and a key prognostic and monitoring marker in these patients, and they remain an important part of the most recent classification criteria [1]. Consequently, understanding potential technical differences between the platforms and assays used by laboratories today to measure these antibodies is essential, and thorough validation of novel diagnostic methods is mandatory before they are introduced in routine clinical practice. Furthermore, the standardization of anti-dsDNA antibody assays remains challenging [5].

Historically, Farr RIA was regarded as the gold standard for detecting anti-dsDNA antibodies due to its good combination of sensitivity and specificity [5]. However, this assay has been almost entirely phased out of routine use due to radiation exposure, and for this reason, it was excluded from our study. In addition, the most recent reports of independent external quality assessment programs, such as the College of American Pathologists (CAP) Survey for the USA and UK National External Quality Assessment Service (NEQAS) for other regions, have revealed the current anti-dsDNA assay landscape, and in the dsDNA testing scheme of the CAP Survey for January 2023, no results for Farr RIA were listed. Instead, 183 participants used ALBIA, 152 used ELISA, 64 used FEIA, 49 used CIA, and 11 used immunoblots (as summarized in CAP’s September 2023 report for ‘Anti-DNA Antibody Double-Stranded (ds)’) [18]. Similarly, in UK NEQAS for January 2023, only 9 participants used Farr RIA, while 315 used enzyme immunoassays (including both ELISAs and FEIA), 90 used CIA, 40 used ALBIA, and 13 used immunoblots (UK NEQAS ‘dsDNA responses for sample 241-1′ of January 2024) [19]. These results confirm that Farr RIA has practically disappeared from the laboratories, and that users have switched to non-radioactive methods.

In order to highlight the evolution of the usage of different anti-dsDNA methods, the data from the UK NEQAS survey covering 2009 to 2017 was used to estimate the distribution of anti-dsDNA antibody methods in the laboratories (Figure 5). The analysis showed a significant increase in the adoption of CIA technology, a decline in ELISA usage, and stability among other methods. This shift can likely be attributed to two factors: the rising demand for fully automated solutions due to the need for efficiency and cost pressure in the laboratories and the robust technical performance of CIA assays.

ELISAs, FEIA, CIA, ALBIA, and more recently, PMAT, made available to laboratories a couple of years ago, have gained popularity due to their convenience through full automation, improved safety by eliminating the use of radioactivity, and, in the case of the multiplex assays, their capacity to detect multiple autoantibodies simultaneously. Furthermore, these alternative assays report quantitative results and are often used alongside or in place of CLIFT, depending on the specific clinical context and laboratory capabilities. Driven by the need to automate antibody testing, fully automated systems are increasingly used for the detection of anti-dsDNA antibodies, as shown in proficiency testing schemes [5].

It should be noted that assays from different manufacturers, even if based on the same methodology (e.g., ELISA), do not necessarily generate identical results. This variability may be due to differences in antigen sources (native vs. synthetic), assay design, or result interpretation, even if the assays are fully automated [13]. External quality assessment schemes consistently demonstrate such variability.

### 4.1. Correlation of Different Anti-dsDNA Assays

When the correlation between the anti-dsDNA assays used in this study was analyzed, an overall relatively high degree of correlation was observed between most of the assays. In concordance with the study by Infantino et al. [20], a high level of correlation was found between PMAT and CIA (0.94), both bead-based assays that use a synthetic anti-dsDNA antigen, followed by CIA and ALBIA (0.84) and PMAT and ELISA (0.81). The lowest correlation was found between ALBIA and CLIFT (0.65), which may be caused by a different antigen used in ALBIA and the different assay design.

The high clinical specificity of the CLIFT used in our study was confirmed by the study of Infantino et al. [20], where it reached a value of 98.2%. Interestingly, this study also used two of the assays used in our investigation, CIA and PMAT, and the comparison based on clinically defined samples showed specificities of 87.5% and 96.4% for those assays, respectively. The suggested high specificities of both PMAT and CIA are confirmed by clinical studies from Bizzaro et al. [16], showing a value of 93.7%, and Infantino et al. [6], resulting in clinical specificities of 96.4% for PMAT and 87.5% for CIA while CLIFT had a value of 98.2%.

### 4.2. Performance Relative to CLIFT

For the diagnosis of SLE, it has been proposed that one can use a screening assay with high sensitivity (detects both high- and low-avidity antibodies, e.g., ELISA or FEIA, or CIA) followed by an assay with high specificity (e.g., CLIFT). Even though the performance of CLIFT is known to have certain limitations associated with its low sensitivity and semi-quantitative nature, its clinical utility for differentiation between SLE and non-SLE patients is widely recognized. Given its high specificity, CLIFT is often used as a confirmatory assay after a positive first-line test result, generated by a different method, such as the ones used in our study. Therefore, it is highly advantageous for the interpretation of diagnostic results if the first-line test closely aligns with the performance characteristics of CLIFT, thereby minimizing the occurrence of discrepant results.

Although CLIFT is often used as confirmatory test for anti-dsDNA antibodies, performing two assays is not always feasible for laboratories due to cost, workflows, the need for automation, and/or reimbursement constraints. Consequently, alternative approaches based on combinations of methods are required to provide both high sensitivity for screening and high specificity for confirmation. When CLIFT was compared to the relative diagnostic performance of the assays ELISA, CIA, and PMAT, very high AUC values were shown (0.95, 0.94, and 0.93, respectively). This demonstrates good agreement of all three assays with CLIFT and would position them as an alternative for anti-dsDNA measurement when CLIFT is not possible or skipped due to the reasons mentioned above. ALBIA showed a lower AUC of 0.86, which may be explained by a different assay design.

Additionally, an approach based on fixing the specificity was used to further investigate how the assays used in this study compared. This analysis allows for a more objective comparison of additional performance characteristics, and when a standardized threshold is applied, the sensitivities can be directly compared [13]. When the specificity of all assays was set to 94.4%, the sensitivities ranged from 84.7% for CIA and ELISA, followed by 76.3% for PMAT to 42.4% for ALBIA.

Although there is no single method that has both high sensitivity and specificity, these results reveal that ELISA, CIA, and PMAT represent useful first-line tests, given their performance characteristics and their high level of agreement with CLIFT. Furthermore, these assays are quantitative, a valuable feature considering that antibody levels may provide additional clinical insight [21]. In this cohort, ALBIA demonstrated a significantly lower diagnostic performance compared to the other methods, suggesting its limited utility—particularly when used alone without CLIFT confirmation.

### 4.3. Correlation Between Anti-dsDNA Assays and C3 as Proxy for Disease Activity

Anti-dsDNA antibody assays are not only used for the diagnosis and classification of SLE but also for the monitoring of disease activity and assessing the risk of lupus nephritis (LN). While some assays are particularly useful for the diagnosis of SLE, others might be superior to assess active disease (flares) [22,23]. A decrease in C3 levels is often associated with increased disease activity in lupus. When the disease flares up, it can lead to the consumption and depletion of complement proteins like C3, as they are involved in immune response. Several studies investigated the inverse correlation between the C3 level and disease activity. Zhang et al. [24] found a moderate discrimination to predict SLE disease activity, while the study of Shang et al. [25] resulted in the same correlation between SLEDAI or C3 and anti-dsDNA antibodies. A more recent publication investigated 175 sera from 99 patients with SLE, 31 patients with other connective tissue diseases (CTD), and 20 healthy individuals (HI) [26]. The samples were tested for antibodies against dsDNA, C1q, nucleosomes, and histones and for C3 and C4 complement components. Multiple linear regression analysis demonstrated that disease activity in SLE could be predicted by the levels of antibodies against dsDNA determined by standard (*p* < 0.05) and HA ELISA (*p* < 0.001) and inversely associated with the concentration of C3 (*p* < 0.001). Thus, the C3 level seems to be a good surrogate indicator of disease activity in the absence of clinical data.

Although C3 data was only available for 49 of the patients included in the study, the results showed a negative correlation between C3 and anti-dsDNA measured by the different assays and different levels of correlation between C3 and anti-dsDNA levels based on the assay used, with the highest correlation for the ELISA (0.81), followed by CLIFT (0.76), CIA (0.70), PMAT (0.70), and finally ALBIA (0.51). An inverse association with C4 was also observed, however, with lower correlation coefficient values compared to that of C4.

The main limitations of our study were the lack of available clinical data (specifically SLE diagnoses) for the included patients, as well as the limited number of patients with available complement C3 and C4 data. However, a significant strength lies in the number and variety of anti-dsDNA methods and platforms employed, which accurately represent the reality of real-world laboratory settings. In addition, the samples in this study originate from routine testing at a large U.S. reference laboratory, which gives a perspective of a real-world clinical testing environment, as opposed to a controlled study design with predefined cases and controls. This approach enhances the generalizability of our findings by capturing a broader and more diverse patient population, including individuals at varying disease stages and with heterogeneous clinical presentations—conditions that are often challenging to assess across different immunological methods. This allows for the evaluation of test performance and biomarker patterns under conditions that mirror actual clinical practice, thereby increasing the relevance and potential use of the results. Lastly, this study provides valuable insights into the utilization and application of various anti-dsDNA diagnostic techniques, highlighting the importance of understanding their technical differences in the context of their use for screening versus confirmatory purposes and addressing scenarios where the use of multiple methods may not be feasible.

## 5. Conclusions

Anti-dsDNA antibodies measured across different methods showed varying diagnostic performance and agreement with each other and with CLIFT. Overall, ELISA, CIA, and the novel PMAT showed very high correlation to each other and to CLIFT, establishing them as good first-line tests for SLE screening followed by CLIFT for confirmation, and even in the absence of CLIFT. In contrast, in this cohort, ALBIA revealed lower clinical performance and correlation with CLIFT, suggesting that it might not be suited as an alternative for CLIFT. The agreement between anti-dsDNA and complement consumption, used as a surrogate for disease activity, also exhibited high variation, with ELISA, CIA, and PMAT showing strong concordance with low C3 levels, while the correlation for ALBIA was significantly lower, suggesting that the first three assays may be more effective in assessing and monitoring disease activity.

## Figures and Tables

**Figure 1 diagnostics-15-02430-f001:**
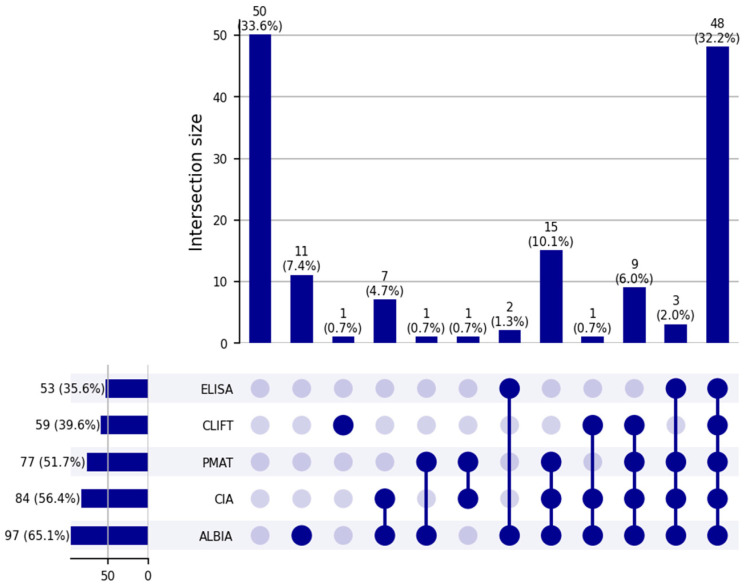
UpSet plot demonstrating the overlap between anti-dsDNA antibodies measured using different methods.

**Figure 2 diagnostics-15-02430-f002:**
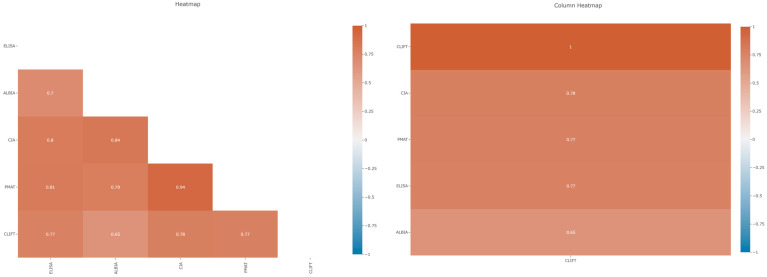
Left panel: heatmap showing the correlation between five anti-dsDNA antibody assays. The highest correlation was found for CIA vs. PMAT and the lowest correlation for ALBIA vs. CLIFT. Right panel: column heatmap showing relative correlations of four assays to CLIFT, which was set to 1.00. The highest correlation with CLIFT was found for CIA, closely followed by PMAT and ELISA, and then ALBIA.

**Figure 3 diagnostics-15-02430-f003:**
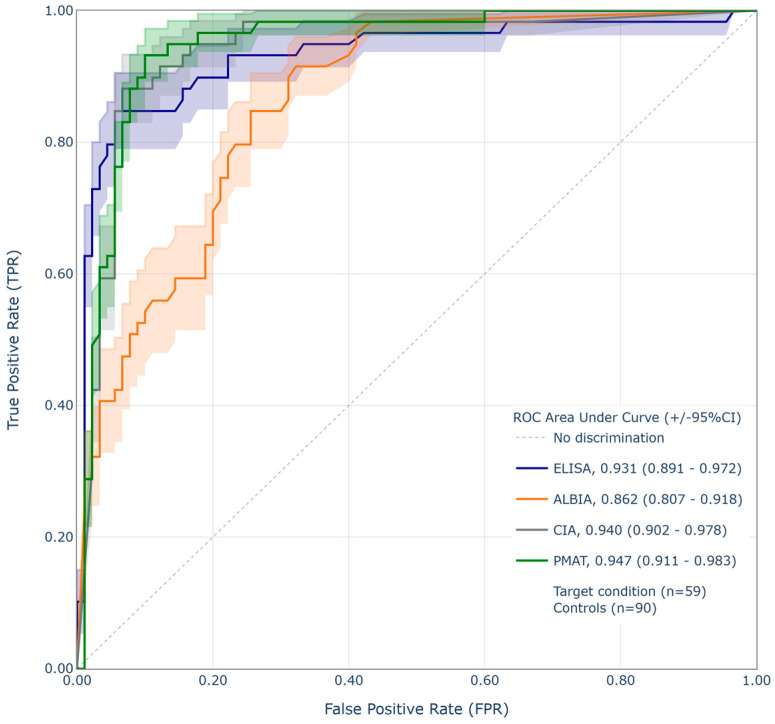
Receiver operating characteristic (ROC) analysis of the four immunoassays (ELISA, ALBIA, CIA, and PMAT) using CLIFT as a binary classifier.

**Figure 4 diagnostics-15-02430-f004:**
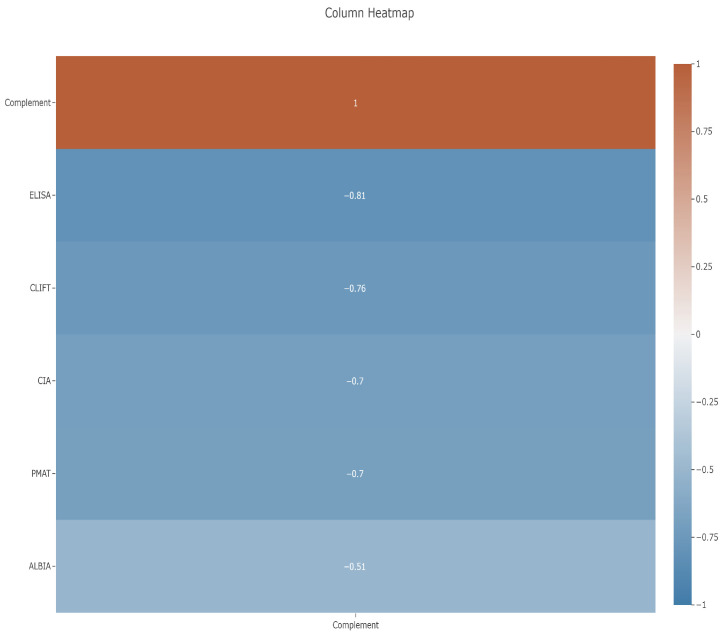
Column heatmap showing relative correlations of five assays to C3, which was set to a value of 1.00. The highest correlation was found for ELISA, followed by CLIFT, CIA, PMAT, and then ALBIA.

**Figure 5 diagnostics-15-02430-f005:**
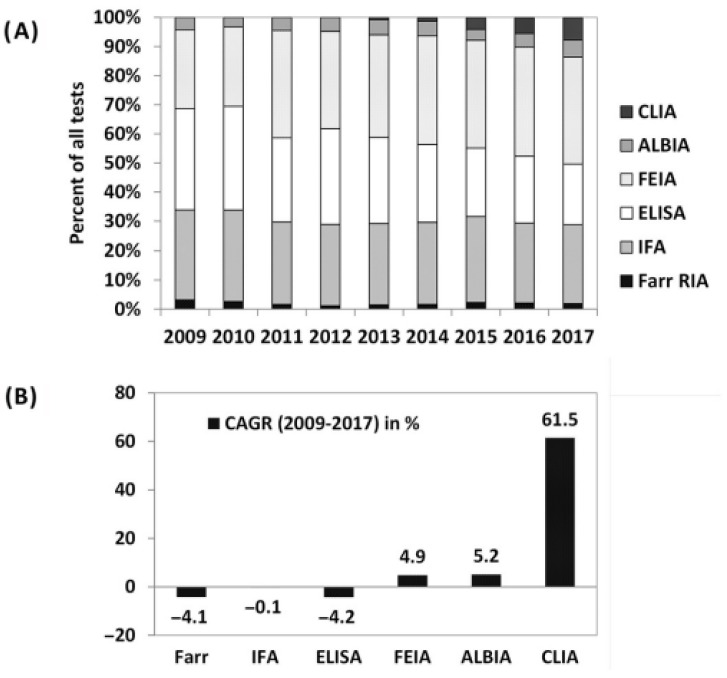
Graphics showing a change in usage of methods for anti-dsDNA testing for the years 2009–2017 based on the results of the UKNEQAS results for anti-dsDNA sample distributions [5]. In (**A**) the relative proportion of the different methods is shown (from 2009). In (**B**) the compound average growth rates (CAGR, in percent) are displayed [5]. This content is copyright of Werfen—© Werfen 2018 All Rights Reserved. Note: Different nomenclature/abbreviations were used for the following assays in this figure: IFA = immunofluorescence assay; CLIA = chemiluminescence immunoassay.

**Table 1 diagnostics-15-02430-t001:** Technical characteristics overview of the anti-dsDNA assays used in this study.

Technology	CLIFT	ELISA	ALBIA	CIA	PMAT
**Platform**	EUROPattern	QUANTA Lite HA	BioPlex 2200	QUANTA Flash	Aptiva
**Manufacturer**	Euroimmun	Werfen	Bio-Rad	Werfen	Werfen
**Assay time**	90 min	90 min	45 min	30 min	30 min
**Detection**	Semi-quantitative	Quantitative			
**Analytical measuring range**	N/A	12.3–1000 IU/mL	1–300 IU/mL	9.8–666.9 IU/mL	2.30–814.10 IU/mL
**Cut-off value**	N/A	≥30 IU/mL	≥5 IU/mL	≥35 IU/mL	>35.00 IU/mL
**Interpretation**	N/A	≤30 negative >30 positive	1–5 negative 5–9 indeterminate >9 positive	9.8–35 negative 35–45 equivocal >45 positive	2.3–27.00 IU/mL negative 27.00–35.00 IU/mL indeterminate >35.00 IU/mL positive
**Solid phase**	Slide	Well	Bead		
**Antigen source**	*Crithidia* *luciliae*	Native calf thymus	Synthetic dsDNA		

**Table 2 diagnostics-15-02430-t002:** Agreement between methods for anti-dsDNA detection based on Spearman’s correlation analysis, showing *ρ* values with 95% confidence intervals.

Spearman	CLIFT	ALBIA	ELISA	CIA	PMAT
**CLIFT**					
**ALBIA**	0.65 (0.54–0.73)				
**ELISA**	0.77 (0.69–0.83)	0.70 (0.61–0.78)			
**CIA**	0.78 (0.71–0.84)	0.84 (0.78–0.86)	0.80 (0.74–0.86)		
**PMAT**	0.77 (0.70–0.83)	0.79 (0.72–0.84)	0.81 (0.75–0.86)	0.94 (0.91–0.96)	

**Table 3 diagnostics-15-02430-t003:** Relative performance of anti-dsDNA assays when compared to CLIFT at a fixed specificity of 94.4%.

Assay	Original Cut-Off	Threshold	Relative Specificity %	Relative Sensitivity %
PMAT	35 IU/mL	25.9 IU/mL	94.4	76.3
ALBIA	10 IU/mL	170 IU/mL	94.4	42.4
ELISA	≤30 IU/mL	70.7 IU/mL	94.4	84.7
CIA	≥35 IU/mL	144.8 IU/mL	94.4	84.7

## Data Availability

Data will be made available for meta-analysis purposes.

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
