# Peer review of "Comparison of Five Assays for the Detection of Anti-dsDNA Antibodies and Their Correlation with Complement Consumption"

_diagnostics, 2025, doi:10.3390/diagnostics15192430_

Round 1
Reviewer 1 Report
Comments and Suggestions for Authors
I appreciate the editors for the opportunity to review the manuscript entitled "Comparison of five assays for the detection of anti-dsDNA antibodies and their correlation with complement consumption." The study addresses an important aspect of autoimmune diagnostics; however, it requires careful analysis and extensive revisions by the authors to enhance scientific rigor. 1. The following points need to be addressed:
- Line 85: Are these samples from patients with SLE? How was this confirmed? Please clarify in the Methods section.
- The information about the kits in the Methods section is unnecessary. Instead, describe the procedure followed for each assay. Additionally, ensure this section is written in the past tense.
- Figure 2: The AUC for Aptive is reported as 0.95 in Line 171 and 0.94 in Line 167. Please correct this discrepancy.
- Figure 3: The purpose of this figure is unclear as it is not explained in the Results section. Please provide an explanation or remove it.
- There is another Figure 3 on page 9. Either explain its relevance or remove it.
- Figure 4 is not mentioned in the text. Ensure it is properly referenced and discussed.
- Results are provided only for C3, but not for C4. Please clarify what happened with C4 in the Results section.
Good
Reviewer 2 Report
Comments and Suggestions for Authors
The authors present an interesting paper about the concordance of different methodologies for anti-dsDNA testing and their association with complement levels. Here you are my comments:
- Introduction: the statement "Monitoring of anti-dsDNA antibody levels is valuable for assessing disease progression and guiding treatment in autoimmune disorders" is misleading since those antibodies are a marker of disease activity only in SLE, not other AID.
- Ref 2 refers to a study that highlights the association of anti-dsDNA IgA with active disease and lupus nephritis, whereas the authors report it as a ref for their limited clinical relevance.
- Methods: you mention 6 methods while you described 7.
- Results: what is the purpose of reporting correlations both in table 2 and fig 1? Furthermore results are different in table and figure: you used different coefficients? Please clarify.
- Numbers reported in the text are different frome the ones found in figure 2, please revise them.
- Figure 3 mentioned in the text is numbered as figure 4 whereas figure 3 is not mentioned in the main text.
- Discussion: I suggest to highlight the need or novelty of your study and stress out the reason why it should be published, the gap, if there is, you are trying to fill, for instance.
Reviewer 3 Report
Comments and Suggestions for Authors
Dear Authors, very interesting manuscript in which you are testing five different methods for detecting anti-dsDNA and correlate with complement levels in patients with SLE. My comments: could you please give us in an introduction more about the importance of the correlation between antibodies against DNA and complement levels? Why both biomarkers could help to the clinician to assess clinical disease activity? Very nice description about your lab diagnostic methods, but who are de 149 people? Could you give us a brief description about your patients? In results section, what about the correlation with C4? This association is very briefly mentioned in discussion. Could you give us more about anti-dsDNA and C4 levels? Overall is a nice paper. Best regards.
Round 2
Reviewer 1 Report
Comments and Suggestions for Authors
Thank you for the opportunity to re-review the manuscript. The authors have addressed most of the previous comments; however, the following points still require attention:
-
Section 2.2: Please specify whether the assays were performed according to the manufacturer’s instructions. If any modifications were made to the protocols, they should be clearly described to ensure reproducibility and robustness of the assays.
-
Figure 4: This figure does not include graphs for C4. In addition, C4 results are not discussed in the discussion section and should be addressed if relevant.
-
Language and Clarity: Several sections of the manuscript are difficult to interpret due to grammatical and syntactical issues. For example, in Section 3.3, the sentence “C3 data were available for 49/149 (32.9%) of the samples, ranging from 15–200 mg/dL” incorrectly suggests that the samples were ranging, rather than the C3 levels. The sentence should be revised for clarity. A thorough language edit is recommended to enhance the overall readability and scientific clarity of the manuscript.
Reviewer 2 Report
Comments and Suggestions for Authors
Thank you for revising the manuscript according to my comments. Yet, there are some main methodological concerns and several pending issues.
Section 2.2 starts with the sentence "All samples were tested for anti-dsDNA using five out of the six methods described above". However you described 7 not 6 methods.
Fig.2 and 4 Numbers are quite small and very difficult to read. Please amend.
Tab. 3 Very hard to read. Please revise colors' choice.
Fig. 4 I do not understand the left panel: since results are different form fig. 2 I assume you are reporting correlation's coefficients for the 48/9 patients with available C3/4 data. If it is as I assume, this panel generates confusion and I suggest to remove it leaving only the right panel.
Fig. 5 uses different abbreviations from the ones used in the main text and other figures/tables. Please homogenize them.
The scarcity of data on C3/4 should be added as a main limitation of your study. Same for the lack of identification of patients from what samples are drawn (SLE? Other AID? Healthy controls?)
Again, you should clearly state the gap of knowledge you are trying to fill and the novelty of your study, especially considering all the methodological limitations (for instance it is not specified if samples are from patients with SLE, healthy controls, other AID, etc). For instance, these techniques have never been compared?
Round 3
Reviewer 2 Report
Comments and Suggestions for Authors.